# Regulation of Metastatic Tumor Dormancy and Emerging Opportunities for Therapeutic Intervention

**DOI:** 10.3390/ijms232213931

**Published:** 2022-11-11

**Authors:** Vasilia Tamamouna, Evangelia Pavlou, Christiana M. Neophytou, Panagiotis Papageorgis, Paul Costeas

**Affiliations:** 1The Centre for the Study of Haematological and Other Malignancies, Nicandrou Papamina Avenue, 15, Nicosia 2032, Cyprus; 2Karaiskakio Foundation, Nicandrou Papamina Avenue, 15, Nicosia 2032, Cyprus; 3Tumor Microenvironment, Metastasis and Experimental Therapeutics Laboratory, Basic and Translational Cancer Research Center, Department of Life Sciences, European University Cyprus, Nicosia 2404, Cyprus

**Keywords:** metastasis, dormancy, intrinsic mechanisms, tumor microenvironment, escape from dormancy, therapeutic interventions

## Abstract

Cancer recurrence and metastasis, following successful treatment, constitutes a critical threat in clinical oncology and are the leading causes of death amongst cancer patients. This phenomenon is largely attributed to metastatic tumor dormancy, a rate-limiting stage during cancer progression, in which disseminated cancer cells remain in a viable, yet not proliferating state for a prolonged period. Dormant cancer cells are characterized by their entry into cell cycle arrest and survival in a quiescence state to adapt to their new microenvironment through the acquisition of mutations and epigenetic modifications, rendering them resistant to anti-cancer treatment and immune surveillance. Under favorable conditions, disseminated dormant tumor cells ‘re-awake’, resume their proliferation and thus colonize distant sites. Due to their rarity, detection of dormant cells using current diagnostic tools is challenging and, thus, therapeutic targets are hard to be identified. Therefore, unraveling the underlying mechanisms required for keeping disseminating tumor cells dormant, along with signals that stimulate their “re-awakening” are crucial for the discovery of novel pharmacological treatments. In this review, we shed light into the main mechanisms that control dormancy induction and escape as well as emerging therapeutic strategies for the eradication of metastatic dormant cells, including dormancy maintenance, direct targeting of dormant cells and re-awakening dormant cells. Studies on the ability of the metastatic cancer cells to cease proliferation and survive in a quiescent state before re-initiating proliferation and colonization years after successful treatment, will pave the way toward developing innovative therapeutic strategies against dormancy-mediated metastatic outgrowth.

## 1. Introduction

Cancer progression largely comprises the processes of primary tumor growth and secondary metastasis. More specifically, it refers to the phenotypic changes in already-formed neoplastic lesions, including morphological, molecular, and functional changes in tumor cells while incorporating an extensive variability of signals and mechanisms [1] Importantly, a predominant cause of cancer-related deaths is attributed to the emergence of tumor relapse, tumor escape from dormancy, and metastasis, which can occur years or even decades following successful completion of treatment [2,3]. Tumor dormancy is defined as a temporary mitotic G0-G1 arrest and a growth arrest of a dormant cancer mass where the number of cancer cells remains constant by the equilibrium between cell division and apoptosis (Figure 1) [4,5,6,7,8,9]. Dormant cancer cells experience various conditions that hinder their growth, such as chemotherapeutic stimuli, nutritional deficiencies and hypoxia [10]. A state of cell proliferation or dormancy is then determined based on the secretion of factors that regulate various signaling pathways, regulating their growth through interactions with their tumor microenvironment (TME).

Regarding the metastatic potential of cancer cells, there are various molecular and cellular mechanisms underlying the different steps of the metastatic cascade, including the epithelial-to-mesenchymal transition (EMT), invasion, anoikis, transport through vessels, the outgrowth of secondary tumors, and angiogenesis through neovascularization [11]. Neovascularization of tumors is naturally leaking, a feature that would permit not only the passage of tumor cells into the circulation, but also their entry from the circulation back into the tumor. In particular, the invasion–metastasis cascade involves a series of steps, starting with local, primary tumor invasion, leading to cell proliferation and survival in a target organ following intravasation (invasion of a tumor cell through a basement membrane to a lymphatic of blood vessel), circulation survival and extravasation (leakage from a blood vessel or tube into the tissue around it). However, only a subgroup of cancer cells manages to reach a distant organ target, known as disseminating tumor cells (DTCs) (Figure 1). DTCs are detected in the bone marrow and their majority remain in a quiescence state. A subgroup of them are circulated in the blood and they are known as circulating tumor cells (CTCs) (Figure 1), whereas another population of DTCs holds a stem cell-like phenotype known as cancer stem cells (CSCs) [12,13]. Following intravasation into the lymphatic system or the surrounding vasculature, CTCs meet environmental hazards inflicted by the immune and circulatory system; however, through platelet interactions a small proportion manages to suppress the danger [14]. Entry into a proliferative or dormant state is determined during extravasation, following CTCs adhesion to endothelial cells [15].

There are some intriguing similarities between the concept of cancer stem cell (CSC) theory and cancer cell dormancy, including their contribution to disease relapse due to their drug-resistant properties. However, they have some key differences. Dormant cancer cells are known to enter a quiescent state characterized by a reversible, cell cycle growth arrest, while CSCs growth is only delayed in the cell cycle [16]. Asymmetric cell divisions allow for the self-renewal of CSCs, whereas dormant cancer cells exist in identical cell differentiation stages and are able to oscillate between proliferation and dormancy (Figure 1) [17]. Furthermore, the gene expression profile is not shared between dormant cells and their parental lines, the former expressing certain genes that are important for the initiation and maintenance of dormancy.

Additionally, DTCs enter metastatic dormancy in order to adapt to either the bloodstream, the TME, or the target organ (Figure 1) [18]. Therapy-induced dormancy occurs as cancer cells aim to escape anti-cancer treatment, cytotoxic effects, which often target proliferating cells [19]. Clonal cell population heterogeneity also contributes to the production of metastatic phenotypic variants. Another factor determining tumor fate is the duration and dose of treatment leading to dormancy, senescence or apoptosis [20]. However, senescence might enable the acquisition of self-renewal properties and the transformation from a dormant to a proliferative phenotype again [21]. Indirect effects of anti-cancer treatments promoting dormancy also include the generation of reactive oxygen species (ROS), nutrient deprivation and hypoxia, environmental stressors which have direct effects on tumor cells (Figure 2) [22].

In respond to microenvironmental cues, DTCs can re-enter the cell cycle and adjust to their new microenvironment. Thus, it is of paramount importance to understand the biology of DTCs reawakening in order to prevent metastatic outgrowth. In this article, we aim to comprehensively describe dormancy and escape of dormancy mechanisms and explore how the dormant disseminated cells, which are widely understood to be the seeds of metastasis, could represent an effective approach towards limiting the expansion of secondary lesions, as well as how targeting this stage of metastatic spreading may provide a potential therapeutic approach.

## 2. Mechanisms of Dormancy Induction and Maintenance

Dormancy and its maintenance are complex processes, mediated by intrinsic and autocrine signaling, and it is also controlled by signals derived from immune cells, endothelial cells, fibroblasts and other non-cellular components of the tumor microenvironment [23,24]. In this section, we describe the current knowledge regarding the cancer cell-autonomous mechanisms (genetic autophagy-related and epigenetic alterations) that either sustain tumor dormancy or promote the escape from this inactive state (Figure 2). Furthermore, we focus on the extrinsic components controlling this process, which include the role of the microenvironment, with emphasis on the effects of the extracellular matrix (ECM), immune surveillance and angiogenesis in regulating dormancy during colonization to various organs (Table 1).

### 2.1. Intrinsic Mechanisms

DTCs moving from the primary site to a distant site remain in a dormant state by escaping the proliferative cycle or by reaching an equilibrium between proliferation and apoptosis. While being in a dormant stage, dormant DTCs can accumulate additional molecular, epigenetic and genetic aberrations that enable them to optimally adjust in their new microenvironment. The intrinsic mechanisms described in this review refer to the genetic and epigenetic, apoptotic, proliferative, survival and autophagy-related alterations and intracellular signaling (Figure 2).

#### 2.1.1. Genetic Alterations

A correlation has been demonstrated between various alterations in the genome, primarily linked to cell differentiation/proliferation and the induction and maintenance of the dormant phenotype as shown in several preclinical cancer models. Overexpression of the FBXW7 gene has been rendered a crucial component for the maintenance of breast cancer dormancy. In the presence of FBXW7, entrance to the cell cycle is restrained through the ubiquitination and subsequent proteasomal degradation of its inducers c-Myc and cyclin E. When genetically ablated, the FBXW7 gene led to not only the disruption of the quiescent state of DTCs, but also the initiation of their proliferation in both allograft and xenograft models. DTCs lacking FBXW7 were significantly depleted following paclitaxel treatment, thus indicating increased sensitivity to chemotherapy. Poorer prognosis and shorter cell survival was associated with tumors expressing high levels of FBXW7 compared to FBXW7-deficient cells following tumor resection and chemotherapy treatment [25].

A dormant phenotype of breast DTCs in the bones has been linked to the leukemia inhibitory factor receptor (LIRF) which interacts with the interleukin-6 (IL-6) cytokine family. LIRF acts as a suppressor of cytokine signaling (SOCs) and an activator of signal transducer and activator transcription 3 (STAT3). Upon LIFR ablation, downregulation of dormancy-associated genes, reactivation of proliferation and bone colonization was observed [26]. Another gene associated with the maintenance of the dormant phenotype in head and neck squamous cell carcinoma (HNSCC) is the paired-related homeobox transcription factor (PRRX1). Its expression drives downregulation of mIR-642b-3p, which participates in tumorigenesis and cell growth and, therefore, sustains dormancy through p38 and transforming growth factor-β2 (TGFβ-2). Its implication with metastasis and invasion through the activation of the epithelial-to-mesenchymal transition program (EMT) indicates that this process could be exploited to sustain cancer cells in the dormant state [27]. Intraperitoneal and pulmonary metastasis is inhibited by kisspeptin 1 (KISS1), as shown in xenograft models of several cancer types, such as ovarian, breast and melanoma cancers [28]. The reverse role of protein kinase C (PKC) in re-activating cell migration, following inhibition by KISS1, is indicative of its potential as a molecular target for treatment [29]. In summary, genetic alterations play a crucial role in regulating the shift from quiescence to proliferation. Such genetic modifications have been found to not only reactivate cells but also to keep them in a suppressed steady state.

#### 2.1.2. Mechanisms of Mammalian Autophagy in Cancer Dormancy

Autophagy is another proposed crucial mechanism for the re-activation, survival and adaptation of dormant DTCs. Autophagy is an evolutionary conserved, physiological mechanism of fitness and cell survival, whose activation highly depends on the presence of unfavorable, metabolic stress conditions, such as nutrient deprivation. Its mechanism of action is associated with the degradation and subsequent recycling of damaged cytosol components, misfolded proteins, organelles and other macromolecules in order to sustain homeostasis and diminish cell damage. As such, evidence begins to accumulate regarding the exploitation of autophagy by tumor cells to endorse their survival under oxidative stress, improve their bioenergetics and thus facilitate tumor progression (Figure 2) [30]. Proliferating cells generally show decreased levels of autophagy compared to dormant cells.

Furthermore, an association has been made between the induction of a reversible dormancy state through activation of autophagy and the expression of the Aplasia Ras homolog member 1 (ARHI1) tumor suppressor gene in xenograft ovarian tumors. Tumor growth escalated following ARHI1 inhibition, suggesting a role in maintaining tumor dormancy. ARHI1 promotes autophagy through the upregulation of ATG4 cysteine protease and inhibition of the mammalian target of rapamycin and phosphatidylinositol 3-kinase-protein signaling pathway (PI_3_K/AKT/mTOR) [31]. It further enhances autophagy by driving nuclear localization of Forkhead box O3 (FOXO3a) and transcription factor EB (TFEB), which are pivotal mediators for the expression of various autophagy effectors, such as the microtubule-associated proteins 1A/1B light chain 3B (MAPLC3B) [22]. Contradicting results were acquired regarding the consequences of ARHI1 expression in vivo and in vitro, which led to cell dormancy and autophagic cell death, respectively. This is potentially attributed to survival signals induced by insulin growth factor 1 (IGF-1), interleukin 8 (IL-8) and vascular endothelial growth factor (VEGF) and their ability to release ovarian cancer cells from autophagic cell death in vitro. In contrast, these signals increased survival through delaying the outgrowth of the dormant ovarian tumor xenograft mouse models.

The highly active status of autophagy in dormant DTCs has further been verified by numerous studies, analyzing cancer cells derived from diverse types of cancer, including bone, gastrointestinal and pancreatic cancers. The expression of insulin-like growth factor 2 (IGF2) has been shown to induce a state of dormancy in osteosarcoma, leading to acquired resistance against chemotherapeutic drugs and the subsequent, future endangerment of minimal residual disease. Increased levels of autophagy, enhanced cell survival and a downregulation of downstream signaling by IGF were associated with dormant state. The critical role of IGF in cell survival was further shown following inhibition of autophagy that led to significantly increased chemotherapeutic sensitivity [32]. Similarly, in pancreatic duct adenocarcinoma, cell dormancy was induced by copper deprivation, a consequence of many anti-cancer therapies leading to higher autophagy levels. The dormant state of cells was rendered by the simultaneous decrease in ATP production, and increase in mitochondrial ROS (Figure 2) [33]. Taking into account the studies performed to date, there is a strong indication that autophagy is involved in various stages of cancer formation and progression. Moreover, autophagy plays a role in several events leading to tumor cell survival, resistance to treatment and metastasis, emerging as one of the critical factors of the dormant state. Through numerous independent studies using cancer cells derived from a wide range of different cancers, and from mice tumors or xenograft models, autophagy seems to be highly active in dormant cancer cells.

#### 2.1.3. Intracellular Signals

In the case of aggressive cancer development, tumor cells adapt to unique microenvironments by withstanding various cellular stresses. Various cell-derived factors have been proposed to act upon the modulation of signaling pathways between cell dormancy and cell growth. Consequently, intracellular signaling pathways are thought to affect the fate of cancer cells and drive the shift between their state of dormancy and tumorigenesis. The first signaling mechanism that has been associated with the dormant phenotype in many preclinical models involves the balance between p38α/β and the extracellular signal-regulated kinases (ERK1/2), part of the mitogen-activated protein kinase signaling pathway (MAPK) (Figure 2) [34]. In fact, higher ratios were proposed to be analogous to tumor growth induction whereas lower ratios enhance tumor dormancy, with urokinase plasminogen activator receptor (uPAR) being a crucial modulator of this process [35]. The expression of uPAR in the presence of fibronectin activates both the epidermal growth factor receptor (EGFR) and alpha-5 beta-1 integrin [36]. Some of the mechanisms behind the ERK/p38 pathway, leading cells to a quiescent state, have been proposed following studies in a model of aggressive, human epidermoid-carcinoma cell line (HEp3). The following pathways correspond to cells exhibiting an ERK/p38 low ratio, a key characteristic of cell dormancy (Figure 2) [37]. Cell cycle arrest in the G0-G1 phases and, therefore, entrance to the quiescent state can be regulated by overexpression of transcription factors p53, BHLHB3, NR2F1 and the downregulation of c-Jun and FOXM1 (Figure 2) [34]. Furthermore, prevention of apoptosis can be accomplished through Bax inhibition via the upregulation of chaperone BiP/Grp78, a process referred to as adaptive survival following chemotherapy or other stressor-inducing factors. Reversal of the p38^high^/ERK^low^ ratio elicits exit from dormancy and a tumor proliferative phenotype [37].

Cancer dormancy can be separated into cellular dormancy and tumor mass dormancy (rate of proliferation rate = apoptosis rate). Many intracellular factors can regulate tumor dormancy. An example of glioblastoma dormancy regulators include the epidermal growth factor receptor (EGFR) and thrombospondin (TSP) [38]. Other types of cancer exhibit recurrence years after treatment and surgical resection; however, depending on the neoplasm, the period of tumor latency significantly differs [39]. An increased likelihood of metastatic relapse at a distant site over a relatively short period of time, as well as increased levels of cell proliferation have been associated with triple-negative and human epidermal growth factor receptor 2 (HER2) breast cancers, when compared to other subtypes, indicative of their aggressive clinical course [40]. Furthermore, cell dormancy has been displayed in estrogen receptor (ER)-positive breast cancer cells, allowing for their survival 5 years following surgical removal of the primary tumor [41]. The shift from cellular dormancy to cell proliferation was also evident through bone TME remodeling by interleukin-6 (IL-6) and the tumor necrosis factor-alpha (TNFα) in metastatic breast cancer cells [42]. Contrastingly, TSP has been associated with cuing metastatic breast cancer cells into dormancy [43]. In prostate cancer, clinically evident recurrence often occurs years after the prostate-specific antigen (PSA) serum levels increase. Crosstalk between the osteoblast-secreted Axl ligand and transforming growth factor beta-2 (TGF-β2) has been shown to contribute to pancreatic cellular dormancy [44]. Recently, the tyrosine kinase receptors AXL, together with MER, were identified to promote a dormancy-to-reactivation axis within melanoma cells [45].

Another signaling pathway that has been investigated for its role in promoting tumor dormancy and the regulation of the metastatic phenotype in estrogen receptor-positive (ER+) breast cancer cells is NFkB. Following the expression of a ‘constitutively active form of IkB kinase b (CA-IKKβ)’ in cell lines, a reversible inhibition of E2-dependent cell proliferation and tumor growth was demonstrated in vitro and in vivo. Similarly, co-activation of IKKβ and ER enhanced cell invasion and migration to guide the experimental development of metastasis [46]. Decreased activity of the phosphatidylinositol 3′-kinase (PI_3_K)/AKT signaling pathway, comprising the extracellular stress protein clusterin and IGF1, under external stimuli is another mechanism thought to be involved in the switching between cancer cell proliferation and dormancy (Figure 2 and Figure 3). The negative regulation of the pathway is primarily caused by the binding inhibition of IGF1 to its receptor and by the secretion of clusterin under serum deprivation conditions. Therefore, the cancer cell fate between dormancy and proliferation during tumor progression could be dictated by the interplay between IGF-1 and clusterin [47]. The PI_3_K/AKT pathway can also be differentially modulated by the epidermal growth factor receptor (EGFR) under serum deprived conditions (Figure 2). Cell proliferation is stimulated by cyclin-D induction through serum-dependent activation of the EGFR pathway. In the absence of serum, cyclin-D failed to be induced, further supporting the role of EGFR in regulating tumor dormancy and growth [48]. Since the life cycle of dormant cancer cells is affected by intracellular factors, fine-tuning the intracellular pathways can play a pivotal role in the regulation of metastatic dormancy. Due to intracellular activation of many pro-dormancy signaling pathways, tumor cells could transform to a quiescent state and adapt to survive. Multiple lines of evidence indicate that the combination of different intracellular signals can become a vital driving force during dormancy processes.

#### 2.1.4. Epigenetic Mechanisms

Fluctuations between a proliferating and dormant state of DTCs could also be the consequence of epigenetic reprogramming mechanisms, encompassing transcriptional regulation through changes in chromatin structure (Figure 2). Some of those mechanisms include non-coding RNAs, whereas the majority of functions are exerted through histone modifications and DNA methylation [49]. In contrast to its hypermethylation-promoted downregulation observed in numerous cancers, overexpression of the orphan nuclear receptor (NR2F1) has been linked to dormancy of DTCs in prostate and HNSCC cancer patients. Cell quiescence activated by NR2F1 is controlled by the combination of cyclin-dependent kinase (CDK) inhibitors (Figure 3), retinoic acid receptor β (RARβ) and the SOX9 transcription factor. Chromatin repression is further enhanced by NR2F1 through the activation of NANOG, which is involved in the dormant state of DTCs within the bone marrow. Furthermore, RARβ and NR2F1-directed histone H3 deacetylation also contributes to cell dormancy. Therefore, the integration of epigenetic programs of DTCs survival and quiescence through the presence of NR2F1 has been shown to play a pivotal role during both initiation and maintenance of dormancy [50]. Recently, an agonist of the nuclear receptor NR2F1 was discovered. This agonist activates dormancy programs in malignant cells and allows a self-regulated increase in NR2F1 mRNA and protein and downstream transcription of an innovative dormancy program that allowed growth arrest of a human cell lines, an HNSCC PDX line, and patient-derived organoids in 3D cultures and in vivo [51].

In ER^+^ breast cancer patients, the MSK1 kinase has been identified as another crucial regulator of metastatic dormancy, since its higher expression levels correspond to lower probability of developing early metastases. Breast cancer cell differentiation is impaired through the downregulation of MSK1, therefore enhancing their growth potential and bone metastases. The expression of the transcription factors FOXA1 and GATA3, which are vital for cell differentiation, is also regulated by MSK1 depending on the alterations of their chromatin status [52].

Conclusively, several mechanisms for epigenetic regulation of gene expression have been reported in dormant cancer cells. Each of these mechanisms hold unique characteristics that ensure reliable control of cell identity and phenotypic plasticity regarding the intrinsic and the extrinsic dormancy. Understanding the dynamic correlation between dormancy and the epigenome will improve our knowledge for the molecular underpinnings of cell plasticity, and thus lead to the identification of new novel targets for the treatment of metastatic cancers.

### 2.2. Tumor Microenvironment

Tumors comprise the ECM and a complex mass of cellular components (non-malignant and malignant cells) which are conjointly referred to as tumor microenvironment (TME). Ectopic, uncontrolled growth is hindered by mechanisms encoded in all adult tissues, therefore, regulatory mechanisms that contribute to dormancy onset and the prevention of DTCs expansion are assumed to be encoded by tumor-naive target organ microenvironments. Initiation and maintenance of tumor dormancy is, hence, highly dependent on the crosstalk between cancer cells and the TME [53].

#### 2.2.1. Extracellular Matrix (ECM)

The interplay between cancer cells and their microenvironment strongly influences tumor progression with respect to tumor metastasis, annihilation or induction of dormancy [54]. P38/ERK ratio plays a catalytic role during switching between the dormant and proliferative state through the interaction with the ECM, which is a crucial TME component (Figure 2) [55]. The simultaneous α5β1 integrin and urokinase-plasminogen activator receptor (uPAR)-driven ERK activation and the fibronectin-mediated p38 suppression spawn the proliferative nature of human epidermoid carcinoma (HEp3) cells [56]. Growth suppression and therefore the induction of a dormant cell phenotype, however, can be induced by any imbalance between this interaction. P38 hindrance alters this balance followed by a subsequent release of dormant cells into a proliferative state (Figure 2). Hypoxia and ER stress could also compel a pro-survival mechanism following weakened folding of proteins (Figure 3) [57]. B1-integrin suppression has also been linked to the induction of dormancy, as shown in in vitro studies, and using in vivo mouse models for breast cancer. The presence of actin-stress fibers and fibronectin is triggered through integrin activation, thus further elucidating the significance of the TME and ECM in promoting cellular dormancy [58]. In addition to dormancy, chemoresistance can be achieved by the integrin-mediated binding of DTCs to the perivascular niche, where the vascular endothelium provides protection from therapy. Chemosensitivity can be increased via suppressing the integrin-mediated interactions between the perivascular niche and DTCs [59]. Signals received from the extracellular matrix are very important because they can either keep cells in an inactive state or drive their progression into metastasis. Signals from the local microenvironment can guide DTCs to remain inactive. ECM proteins can have a determinant role in sustaining DTCs in a dormant state or allowing tumor development. By uncovering the mechanisms that preserve the cells suppressed or allow for their growth, we may be able to design more effective therapeutic approaches in order to keep DTCs in a continuous dormant state or activate apoptotic signals to eradicate them.

#### 2.2.2. Hypoxia and Angiogenesis

Tumor survival depends on angiogenesis, since tumors unable to intravasate are driven either to apoptosis or in a quiescent, dormant state until the acquisition of sufficient additional alterations, allowing them to escape and restart proliferating [60]. An equivalent rate between programmed cell death and cell proliferation determines metastases and the maintenance of tumor dormancy. The indirect increase in apoptotic rate in tumor cells through inhibitors of angiogenesis regulates metastatic growth [20]. A recent study revealed that DTCs have improved levels of retention, they rapidly extravasate and they had better survival after extravasation, compared to experimentally metastasized tumor cells [61]. 

A hypoxic TME, a key feature of many solid tumors, is a result of oxygen depletion and therefore a disrupted homeostasis [62]. Hypoxia has been proposed as a significant driver of tumor dormancy, eliciting a DTC subpopulation to enter the dormant state and enable adaptation and survival for a long period of time [63]. Angiogenesis and neovascularization, on the other hand, are necessary for the survival and growth of tumors. Failure to enter the vasculature causes cells to undergo increased apoptosis rate until achieving an equilibrium with proliferation and enter a dormant state [64].

Endothelial cells (ECs) produce signals that regulate dormancy in cancer cells, with DII4 (Notch ligand) being a prime example. In ECs, DII4 has been associated with the escape of human colorectal or T-cell acute lymphoblastic leukemia (T-ALL) cells from dormancy, consistent with its role in tumor angiogenesis and the positive regulation of the Notch signaling pathway. In contrast to its absence within quiescent tumors, increased expression of DII4 in aggressive tumors provides further evidence for its contribution in tumor dormancy [65]. Activation of Notch 3 by angiogenic factors in ECs, stimulates a tumorigenic phenotype. While Notch 3 consistently displays decreased expression in dormant tumors, phosphorylated p38, a target of the canonical phosphatase MKP-1 controlled by Notch, was found to be present in increased levels within dormant cells. Therefore, tumor dormancy can be regulated by angiogenesis-driven mechanisms, which include the MAPK and Notch pathways [66].

Angiogenesis suppression and the way it influences dormant lung metastases was investigated by Holmgren et al., in 1995. Tumor cell proliferation was not significantly different between the dormant cells and growing metastasis. Dormant metastasis cells exhibited a more than threefold higher incidence of apoptosis. Angiogenic inhibitors that directed the metastatic growth by increasing apoptosis in tumor cells controlled the process. Moreover, the observation that tumor dormancy was associated with increased, circulating angiostatin levels, suggests that angiogenesis inhibitors could be used to regulate the metastatic growth in tumor cells [67].

Thrombospondin (TSP) was shown to exhibit direct impact in the growth of breast cancer cells (BCCs) and, more specifically, in terms of inducing and maintaining their quiescent state. In the presence of sprouting neovasculature, BBC growth was accelerated and the role of TSP was suppressed [68]. Metastatic breast cancer DTCs are shown to occupy the perivascular niche and, more specifically, the microvasculature. An association between the sustainment of dormancy and the production of thrombospondin by the microvasculature has been previously established [69]. On the contrary, periostin and TGF-β1 production leads to an ‘angiogenic switch’ and a shift towards highly vascularized tumors which have the ability to grow and proliferate rapidly [21]. The survival of dormant cells is also established by endothelial contact and perivascular localization. Reduced oxygen levels in the TME are a crucial guide of cells into dormancy [70]. Other endothelial-derived molecules proposed to influence tumor dormancy and tumor angiogenesis include the vascular endothelial growth factor (VEGF) and epoxyeicosatrienoic acids (EETs), the presence of which renders the escape from tumor dormancy feasible [71]. Dormant state activation, as distinguished by low metabolism and a mitotic G0-G1 cell cycle arrest, was observed following the exposure of MDA-MB-231 breast cancer cells to chronic, intermittent hypoxic conditions in vitro. In contrast, culturing the same cells in normal oxygen conditions restored their proliferation rate, indicating the reversible nature of dormancy [72]. Exposure to hypoxic conditions in vivo revealed the identification of dormancy markers, such as p27, DEC2 and NR2F1, as well as an increased frequency of tumor cells entering dormancy [12]. Various, interconnected pathways could participate in cancer dormancy regulation under hypoxic conditions. In an HNSCC model, TGFβ2 signaling pathway upregulation led to the activation of MAPK p38α/β and the induction of a low ERK/p38 signaling ratio (Figure 2). Subsequently, p27, a proliferation inhibitor, along with SHARP1/DEC2, were induced whilst cyclin-dependent kinase 4 (CDK4) was downregulated, resulting in dormant phenotypes [23]. The dormant-state of cancer cells under hypoxic conditions is initiated and maintained by transcription factors, such as the hypoxia-inducible factor (HIF-1α). HIF-1α overexpression contributes to the decrease in energy resources, such as glucose, ATP and oxygen levels, the suppression of which favors tumor cell survival through dormancy [73]. Extracellular matrix adhesion is essential for normal epithelial tissue, function and homeostasis. Detachment of cells from the ECM leads to apoptosis through anoikis through an increase in metabolic stress. Furthermore, cell survival is promoted through the activation of autophagy by rewiring and suppression of AKT signaling which prevents cells from undergoing anoikis and promotes the entrance of DTCs to dormancy [74]. A further assessment regarding the role of dormancy and its correspondence to non-adherent, metastatic cells was achieved using epithelial ovarian cancer cells (EOC). EOCs progress to metastasis formation following their disengagement from the primary tumor into the fluid-filled peritoneal cavity and re-adherence for the formation of secondary lesions. EOCs persist in their non-adherent form during mitotic arrest in the presence of elevated p27 and p130 levels as well as AKT suppression. However, cell transformation from their quiescent to their proliferative state occurs through the reactivation of AKT signaling following seeding to adherent surfaces [75]. A hypoxic microenvironment has an important role in tumor dormancy because, although most of the tumor cells die upon hypoxia, a part of them can adapt and survive for long periods in a dormant quiescent state. This state can be reversible, with tumor cells recovering the ability to self-renew when new blood vessels reach the hypoxic niche. More experimental as well as theoretical evidence is needed to support how tumor hypoxia both in the primary tumor as well as at target organ sites can influence disseminated tumor cells to enter cancer cell dormancy.

#### 2.2.3. Immune System

The immune system is another fundamental regulator for the induction and maintenance of tumor dormancy (Figure 2) [24]. Alterations accumulating in tumor cells can make them genetically unstable and render them resistant to complete recognition and elimination by the immune system components [76]. The immune system has been indirectly linked to tumor dormancy following observations of increased immune cell infiltration in breast cancer patients with DTCs in the bone marrow (Figure 2).

Adaptive Immunity

T cell content and immune cell activation in tumors in the bone marrow of cancer patients was significantly increased compared to healthy individuals, indicating signs of tumor dormancy. In other words, the bone marrow represented a prosperous site for the presence of lethal DTCs in a dormant state with a potential contribution to this by the high presence of T cells [76].

The maintenance of dormancy in a murine B cell lymphoma model (BCL1) required the presence of cytostatic CD8^+^ T-cells, the depletion of which reversed cancer cell dormancy [77]. Similarly, in a mouse melanoma model, cancer cell dormancy in the lung was found to be governed by CD8^+^ T cells, the decrease in which caused visceral metastases and faster outgrowth, providing further evidence that metastatic growth is regulated by the immune system (Figure 2) [78]. The dormant state of metastatic cells can also be maintained by T lymphocytes, through interferon-γ (IFN-γ) production that was shown to be able to arrest tumor cells to the G0/G1 phase [79].

Cellular dormancy could also be influenced by immune evasion. In a model of acute myeloid leukemia, overexpression of the programmed death-ligand 1 (PD-L1) (also known as B7-H1) led to cytotoxic T cell-mediated resistance to killing and thus a potential contribution dormancy [80]. Dormancy could be also potentiated following the entrance of DTCs in immune-privileged stem cell niches where high levels of regulatory T cells could accumulate under specific circumstances (Tregs) as described by [81].

The development of metastatic melanoma was previously reported in cases following kidney transplantation. The donor turned out to be an individual who underwent tumor resection years before, suggesting the presence of dormant, residual melanoma cells within their body prior to dormancy escape and metastatic progression, due to entry in a new TME and immune system [82]. A switch from a naive to a proliferative tumor phase following the latency period of five months was also observed in a sarcoma model. During the former state, the proliferation marker and G0 exit determinant Ki-67, appeared to be expressed at low levels whilst high levels were associated with the apoptotic fraction, suggesting a connection with CD8^+^ T cell mediated cytotoxicity. Escape from dormancy and an increase in proliferation was noted after depletion of CD4^+^ and CD8^+^ T cells in a pancreatic cancer mouse model (RIP-Tag2) [83]. Tumor progression in this model was shown to be inhibited even in the presence of CD4^+^ T cells alone and absence of CD8^+^ T cells, and that was mediated via the tumor necrosis factor receptor 1 (TNFR1) signaling and the interferon-γ (IFN-γ) pathways. Another mechanism of how CD4 T cells could contribute to tumor dormancy was described by another study where they showed that the effects of CD4^+^ T cells were mediated through the secretion of angiogenesis inhibitors CXCL10 and CXCL9 [84]. Livers from mice and patients with pancreatic ductal carcinomas bared DTCs that lacked expression of the major histocompatibility complex I (MHC I) and cytokeratin 19 (CK19) tumor antigen, providing evidence for a novel immune escape pathway which involves the evasion of T cell killing upon recognition under endoplasmic reticulum (ER) stress [85]. 

‘Immunocloaking’ is a theory supporting that cancer cells, in order to avoid destruction by the immune cells, attack back by evolving ‘cloaking’ mechanisms which they can use to hide themselves from the immune system. For example, dormant myeloma cells are inaccurately recognized as CD169^+^ bone marrow macrophages and osteomacs or other myeloid cells, thus avoiding immune surveillance and detection [86]. With respect to ‘immunocloaking’ the inhibitory receptors, Fc receptor-like protein 6 (FCRL6) and lymphocyte activation 3 (LAG3) are captured by MHC II on CD4^+^, CD8^+^ T cells and NK cells, thus achieving inhibition of immunity against cancer cells expressing MHC II [87,88,89]. LAG3 in particular has demonstrated eminent ability in the suppression of cytotoxic T cell killing, protecting melanoma cells from apoptosis and contributing to tumor survival [90].

Innate Immunity

A recent publication studied how the different macrophage lineages contribute to the tumor microenvironment and how they promote human non-small cell lung carcinoma (NSCLC). For this reason, the researchers performed single-cell RNA sequencing of tumor-associated leukocytes. They recognized distinct populations of macrophages in the lung tumors, characterized by different origins and distinct temporal and spatial distribution in the TME. Macrophages were gathered close to tumor cells early during tumor formation to stimulate epithelial–mesenchymal transition and invasiveness in tumor cells, and promoted a potent regulatory T cell response that protected tumor cells from adaptive antitumor immunity [91]. Depletion of tissue-resident macrophages reduced the numbers of regulatory T cells, endorsed the accumulation of CD8^+^ T cells and reduced tumor invasiveness and growth [91]. Evidently, natural killer cells (NKs) have a significant impact in the suppression of spontaneous metastases formation in colon cancer patients by decreasing the circulating tumor cell content and by inducing dormancy in a colon cancer mouse model in a perforin-dependent manner [92].

Poor clinical outcomes and decreased survival have been correlated with the abundance of CCL18 in the blood. Breast cancer cell metastatic seeding is promoted by CCL18 cytokine that can be secreted by tumor-associated macrophages (TAMs) at the secondary organ. It promotes cancer cell invasion by stimulating clustering of integrins and enhances adhesion of cancer cells on lung tissue through its receptor PITPNM3 [93]. A recent study provided additional insights into the role of cytokines in regulating dormancy. Finally, the identification of an interleukin-binding receptor (IL-1R1) expressed in dormant ER^+^ breast cancer cells which exhibit increased aldehyde dehydrogenase levels (ALDH^+^), active IL-1 signaling pathway and resistance to antiestrogen therapy was associated with unsuccessful treatment; hence, inhibition of the IL-1 pathway could be one potential to increasing targeting of this dormant tumor cell subset [94].

The role of the immune system in controlling tumor cell dormancy in primary and metastatic foci has been underlined by several studies. The current discovery of new immune checkpoint inhibitors has marked a new era of hope for cancer therapy. Nevertheless, the development of immunotherapy against dormant cells is puzzling, as quiescent DTCs have developed new unknown mechanisms to evade the immune system. More therapies that mobilize the immune system against dormant DTCs need to be developed. Despite the current studies, there is still a lot of room to clarify the mechanisms involved in immune-mediated induction of dormancy in tumor cells. This includes understanding how tumor evasion and dormancy are coupled or not.

## 3. Escape from Dormancy

Cancer cells originating from a primary tumor mass persist in a clinically undetectable, dormant state for a protracted period of time, following their dissemination to distant sites. This is a consequence of the new, often unfavorable microenvironment they encounter during their residency at distant regions, causing their entry into senescence or dormant state. Dormancy entry is mediated by either a withdrawal from the proliferative cell cycle or the achievement of equilibrium between apoptosis and proliferation [30,95]. In the meantime, the accumulation of genetic and epigenetic aberrations renders DTCs fully adapted to the new microenvironment (Figure 2). A subgroup of DTCs escape their dormant state and ‘re-awake’, moving on to the formation of expanding masses, in response to not yet fully characterized cues [7].

Dormancy escape could be stimulated through modifications in the TME which comprises a heterogeneous population of non-cancer (e.g., pericytes, cancer-associated fibroblasts, tumor-associated macrophages, lymphocytes and leukocytes), cancer cells and the ECM, encompassed by the secretion of numerous molecules, including metabolites, growth factors, cytokines and lymphokines. Through the constantly evolving interaction with TME components, the fate and behavior of cancer cells are dictated depending on the balance between proliferation and quiescence [63]. At the metastatic site, dormant breast cancer cells shift to proliferation following the displacement of type-1 collagen via β1-integrin signaling pathways, also affecting cytoskeletal architecture [96]. The brain, bone marrow and lung microvasculature consist of residing dormant cells. Quiescence is favored by the secretion of the anti-angiogenic factor thrombospondin-1 (TSP-1), however, during neovascularization, this suppressive signal is lost and TSP-1 switches to promoting cancer cell proliferation and significantly accelerating tumor growth rates (Table 1). Therefore, a dormant niche is associated with a stable microvasculature, whereas outgrowing metastases are linked to neovasculature formation. Furthermore, several tumor-promoting factors secreted from the endothelial cells, such as periostin and TGF-β1, were identified as major inducers of new blood vessel formation in head and neck cancer [97].

Moreover, lipid mediators, such as the fatty-acid transporter CD36, can bind collagen or TSP-1 and actively control escape from cancer dormancy, as previously shown in metastatic prostate and oral cancer cells [98,99]. CD36-mediated tumor growth is achieved through lipid oxidation stimulated by the uptake of fatty acids [100]. Along with cytoskeletal re-organization, various elements of the ECM dictate dormancy escape (Figure 3). For instance, activation of periostin drives dormant, metastatic breast cancer cells into a proliferative state [43,68]. Proteomics of the ECM showed that dormant cancer cells accumulate a type III collagen-enriched ECM niche. Tumor-derived type III collagen is necessary to sustain tumor dormancy, as its interference restores tumor cell proliferation through DDR1-mediated STAT1 signaling [101]. Patient samples showed that type III collagen levels were amplified in cancers from patients with lymph node-negative head and neck squamous cell carcinoma in comparison to patient samples positive for lymph node colonization [101]. Dormancy escape was also induced through the overexpression of vascular cell adhesion protein 1 (VCAM-1) in dormant, metastatic breast cancer lines residing in the bone marrow [102] (Table 1). Moreover, myosin light chain kinase (MLCK) inhibition was previously associated with the initiation of the dormant phenotype and the decrease in metastatic breast cancer in the lungs. Following activation of MLCK by β-integrin and fibronectin, a state of proliferation prevailed over dormancy [38]. Micrometastases formation might also be triggered following primary tumor resection [103], a surprising observation possibly attributed to the ablation of primary tumor-secreted endostatin and angiostatin, angiogenesis inhibitors often protecting against metastatic growth at other tissues [104]. On other hand, more recent contradicting evidence suggests that the systemic inflammatory response induced after surgery endorses the appearance of tumors whose growth was elsewise controlled by a tumor-specific T cell response. Such evidence is linking surgery with subsequent wound-healing responses to the outgrowth of tumor cells at distant anatomical sites [105].

Another proposed mechanism involved in tumor escape from dormancy is chronic inflammation of the host tissue (Figure 2). An example is chronic inflammation generated by either smoking cigarettes or from liposaccharides derived from bacteria, both of which drive neutrophil activation and therefore the formation of ‘neutrophil extracellular traps’ (NETs). NETs are crucial for the re-activation of DTC proliferation, as they secrete matrix metalloproteinase 9 (MMP-9) and neutrophil elastase (NE) following neutrophil degradation. Protease secretion amplifies ‘basement membrane laminin-111’ expression, which interacts with the α3β1 integrin (Table 1) [106]. In addition to these, ECM remodeling by proteolytic enzymes and chromatin scaffold formation are also a consequence of NET production, and can regulate tumor cell dormancy [107].

Re-awakening and re-activation of breast DTCs proliferation could also be attributed to monocyte chemoattractant protein-1 (MCP-1), interleukin-8 (IL-8) or the secretion of other soluble factors from the hepatic stellate cells (HSCs), located in the liver, induced via the ERK pathway [108]. When it comes to the bone marrow microenvironment, DTCs exited their dormant state following the overexpression of inflammatory cytokines or upon other alterations within the bone secretome, such as interactions with specialized tissue-specific cells and nutrients/metabolites [109]. Mechanisms that mediate dormant cell re-awakening might differ depending on the metastatic niche they occupy as each microenvironment responds to different challenges. DTCs exist in their dormant state for a protracted period of time in areas that are highly vascularized, such as the bone marrow [110]. Their escape is highly imposed by the activation of osteoclasts [111]. Following osteoclast recruitment and activation by VCAM-1, a chain of events takes place, including tumor expansion and bone destruction [112]. Activation of TGFβ1 or interleukin-8 (IL-8) and interleukin-6 (IL-6) secretion could be activated upon stromal injuries induced in the bone marrow and therefore reactivate dormant breast cancer cells (Figure 3) [113].

Myeloma dormant cells were greatly reduced following resorption of osteoclasts by the receptor-activator nuclear factor-kb ligand (RANKL) [114]. Similarly, the formation of bone metastases initiated from dormant prostate and breast cancer cells was evident upon castration/ovariectomy [115]. Various proliferation-promoting mechanisms have been implicated in the lung with regards to the escape of dormant breast cancer cells. Some of these include initiation of proliferation by the activation of tank-binding kinase-1 (TBK1) [116], as well as the overexpression of periostin, driving Wnt signaling. Depending on the target organ, distinct mechanisms of escape from dormancy are implicated. This is evident by the effects of Coco, a TGFβ1 ligand antagonist, in metastatic sites of the lung, resulting in re-activation of breast cancer cells, but not in brain or bone metastases (Figure 3, Table 1) [117,118]. Lately, WNT5A was also identified as an activator of dormancy in melanoma disseminated cancer cells within the lung. Age-induced reprogramming of lung fibroblasts increases their secretion of the soluble Wnt antagonist sFRP1, which hinders WNT5A in the same cell population and thus allows for the effective metastatic outgrowth [45]. The reawaking of DCCs is due to responses in the intracellular and local environmental signals. These signals are not fully understood, resulting in the recurrence and metastasis. Thus, understanding the biology of DCC reawakening is key to inhibiting metastasis. Over the last years, a growing body of publications has discovered the signals involved in cancer dormancy and reawakening. Cytotoxic activity caused by the immune cells can lead to a cancer dormant state, whereas chronic inflammation can reactivate cancer proliferation at distant sites through the activation of various signaling pathways.

**Table 1 ijms-23-13931-t001:** Factors implicated in the mechanisms sustaining dormancy or escape from dormancy.

Mechanisms that Sustain Dormancy				
Factor	Mechanism	Cancer Type	Metastatic Site	Reference
Fbxw7	Cell cycle control	Breast	Lung	[25]
LIFR	Hypoxia	Breast	Bone Marrow	[26]
PRRX1	EMT	HNSCC	Lymph Nodes	[27]
KISS 1	Hormone Regulation	Melanoma, Breast, Ovarian	Lung, Intraperitoneal sites	[29]
Wnt5a	Development	Prostate	Bone	[45]
IKKβ	Inflammation	Breast	Multiple Sites	[46]
NR2F1/NANOG	Development, Differentiation	Prostate, HNSCC	Bone Marrow	[50]
MSK1	Differentiation	Breast	Bone	[52]
TGF-β2	Development, Morphogenesis	HNSCC	Bone Marrow	[58]
IFN-γ	Immune Response	Sarcoma	Multiple Sites	[79]
**Mechanisms that Promote Escape from Dormancy**				
**Factor**	**Mechanism**	**Cancer Type**	**Metastatic Site**	**Reference**
VEGF-A	Angiogenesis	Melanoma, Lung	Brain	[71]
TSP-1	ECM constituent	Breast	Bone Marrow	[97]
MMP-9	Metabolic Processes	Breast	Brain	[106]
Coco	Morphogenesis	Breast	Lung	[117]
VCAM-1	Cell adhesion	Breast	Lung/Bone	[119]

## 4. Therapeutic Implications of Tumor Dormancy/Treatment Strategies

The likelihood of relapse could be significantly decreased upon application of therapeutic interventions directly targeting tumor dormancy (Figure 3). During cancer progression, dormant tumor cells often co-exist with rapidly proliferating cells, thus contributing to the development of therapeutic resistance [120,121,122]. Conventional therapies almost exclusively target fast-proliferating cells, rendering dormant cells resistant to those treatments. The eradication of tumor cells could be feasible through the identification of novel therapeutic approaches by unraveling the mechanisms responsible for controlling the transformation between proliferation and dormancy (Figure 3) [23]. Mechanisms focusing on differentiating dormant cells from other non-proliferative yet viable cells, through the identification of their distinct characteristics, could be promising. However, despite the evident contribution of dormant cells in promoting metastatic disease relapse, their targeting remains unsuccessful [123].

### 4.1. Maintaining Tumor Dormancy

A core therapeutic strategy that has been proposed aims to permanently sustain the dormant state of cells, maintaining residual cells non-proliferative and inactive (Figure 3). Hormone therapy is a promising cytostatic approach which drives cells into quiescence and proliferation arrest through entrance in a cell cycle G0-G1 phase [123]. Similarly, the use of palbociclib, abemaciclib, ribociclib or other CDK4/6 cell cycle inhibitors, which play crucial roles in G1/S phase transition, could promote cancer cell dormancy maintenance (Figure 3) [124]. Another approach tested in HEp3 cells involves the inhibition of urokinase-type plasminogen activator receptor (uPAR) signaling. Tumorigenesis is induced in the presence of uPAR since it activates the α5β1 integrin that has a pivotal role in ERK activation by initiating an intracellular signaling cascade via Src and focal adhesion kinase (FAK) [125]. The activities of Src kinase and ERK, both activated when uPAR is overexpressed, can also be inhibited to prevent the formation of metastatic lesions or avoid dormant cell proliferating outbreaks by promoting translocation of cyclin-dependent kinase inhibitor p27 in the nucleus [126] (Figure 3). Dormant breast DTCs were maintained with both PP1 (Src inhibitor) and U0126 (ERK inhibitor) (Figure 3) [38,127].

A recent strategy to induce the maintenance of dormancy was also assessed in HNSCC using an all-trans retinoic acid and 5-azacytidine-based therapy, resulting in upregulation of NR2F1, DYRK1A and p38 MAPK dormancy factors (Figure 3) [50]. Moreover, a peculiar mechanism of action has been associated with the itraconazole response of dormant colorectal cancer cells. The Wnt pathway was inhibited by itraconazole via the non-canonical Hedgehog signaling pathway. Following the initial proliferative burst induced by the treatment, dormant cells entered an irreversible cell cycle arrest and senescence [128]. This therapeutic approach, however, also has its drawbacks as it inactivates cancer cells without driving them to cell death. It requires life-long therapy which is accompanied with numerous challenges, such as patient compliance, increased toxicity, side effects, increased cost, as well as the possibility of cancer cells acquiring resistance. The effectiveness of maintaining long-term dormancy is also of concern as a subgroup of cells may not respond to treatment due to heterogeneity. For instance, treatment with interferon β (IFN-β) resulted in the cell cycle arrest of only 70% of melanoma cells in mice. Additionally, tumor growth may be possible because of the slow-cycling rate of some dormant cells [129]. It appears that treating dormant DTCs by keeping them asleep is attractive in many ways. Most attractive is that it could avoid the need for chemotherapy, and all of its related side effects, in certain patient populations.

### 4.2. Re-Awakening and Sensitization of Dormant Cells to Therapy

Another therapeutic approach focuses on increasing susceptibility of dormant cells to anti-cancer treatment by disturbing their dormant state and ‘re-awakening’ them (Figure 3). Dormant cell re-entrance into the G2-M phase of the cell cycle would therefore lead to increased sensitivity towards cytotoxic therapy [130]. Reactivation of the cell cycle in dormant cells can be achieved by targeting microenvironment components that promote a dormant phenotype. Osteopontin neutralization, which is secreted in the bone marrow by osteoblasts, drove dormant leukemia cells to enter the cell cycle and initiate proliferation (Figure 3). Residual disease was considerably decreased and killing of neoplastic cells was achieved through the synergistic effect of osteopontin inhibition with cytarabine, a chemotherapeutic anti-metabolite [131]. Re-entrance of quiescent leukemia cells into the cell cycle was made possible through the act of granulocyte colony stimulating factor (G-CSF) which enhanced cytarabine cytotoxicity (Figure 3) [132]. Decreased chemoresistance to 5-fluorouracil was similarly induced through treatment with IFNα, resulting in the proliferation of dormant hematopoietic stem cells [133]. Furthermore, inhibiting the APC-CDH1-SKP2-p27/Kip1 signaling pathway could lead to re-awakening of dormant cells. More specifically, harmine-mediated inhibition of DYRK1A kinase terminated dormancy and enhanced the efficacy of the tyrosine kinase inhibitor imatinib in gastrointestinal tumors [134]. Likewise, DYRK1B repression led to the reactivation of the cell cycle in dormant pancreatic cells and to improved response to gemcitabine treatment (Figure 3) [135]. Evidently, dormancy in myeloma cells is maintained through the interaction of cells with Axl tyrosine kinase within the endosteal niche. Axl inhibition was shown to induce re-awakening of myeloma cells and decreased chemoresistance [136]. Integrin inhibitors act in a similar manner and sensitize DTCs to chemotherapy via the vascular cell adhesion molecule-1 (VCAM-1) and the endothelial-derived von Willebrand factor (VWF) [119]. Despite the highly promising approach of reactivating dormant cells to improve cytotoxic drug efficacy, the clinical translation of preclinical evidence suggesting the use of cancer cell proliferation-promoting drugs will be challenging. An association has been made between this therapeutic approach and the high possibility of provoking uncontrolled cell proliferation and an aggressive tumor phenotype [10]. The heterogeneity of DTCs also imposes a challenge as it may lead to a subgroup of cells being unresponsive to the reactivation of the cell cycle, leading to the persistence of residual disease and the expansion of their genetic repertoire driving therapeutic resistance [24]. DCCs can interpret homoeostatic signals from the microenvironment, thereby evading immune surveillance and chemotherapy and also reawake in response to signals, resulting in recurrence and metastasis. Therefore, understanding the biology of DCC reawakening is critically important to preventing metastasis.

### 4.3. Direct Targeting of Dormant Cells

Another therapeutic approach aims at directly targeting dormant cells to reduce the possibility of future disease relapse (Figure 3) [10]. A worse patient survival rate has been previously associated with the suppression of autophagy pathways strictly in dormant breast cancer cells [137]. However, metastatic outgrowth was only minimally affected by the suppression of autophagy in cells that had already switched to a proliferating state. This indicates that administration of autophagy inhibitors is more effective during the tumor latency phase rather than during the cell proliferative phase. Treatment of dormant cells using epigenetic therapies is also being explored through the suppression of epigenetic enzymes, such as histone deacetylases (HDACs) and lysine demethylases (KDMs) (Figure 3) [138]. Persistent, drug-resistant cancer cell survival requires histone 3 (H3) demethylation on lysine 4 (H3K4), catalyzed by a family of histone demethylases referred to as KDM5. A variety of KDM inhibitors currently exist, among which GSK-J4 is the most commonly used, targeting KDM2B [139], KDM5 [140] and KDM6A/B (Figure 3) [141]. Dormant lung and glioblastoma DTCs were eliminated by GSK-J4, following taxane–platinum-based chemotherapy and dasatinib treatment, respectively [142,143]. Another inhibitor exhibiting increased specificity for KDM5 is CPI-455, whose mechanism of action includes an increase in H3K4 trimethylation (H3K4me3) levels and reduced Drug-Tolerant Persister cancer cell (DTPs) numbers in various cancer cell lines that have undergone chemotherapy treatment. Successful DTC killing by CPI-455 was shown in breast, colon, melanoma and lung cancer models [144]. Furthermore, the repression of survival signaling pathways involved in dormancy maintenance (Src signaling) and dormancy escape (ERK1/2-MEK1/2), could efficiently eradicate dormant DTCs. Complete dormant cell apoptosis was achieved following a combinatory administration of AZD6244 (MEK1/2 inhibitor) and AZD0530 (Src inhibitor) (Figure 3) [126]. A recent approach has emerged regarding the annihilation of DTCs, referred to as ferroptosis (Figure 3). ML210 and RSL3 activators of ferroptosis drove DTC death following treatment with kinase inhibitors, such as the combination of carboplatin (ovarian) with paclitaxel (ovarian, breast, and lung cancer, Kaposi’s sarcoma), erlotinib (non-small cell lung cancer (NSCLC), advanced pancreatic cancer), vemurafenib (melanoma skin cancer) and lapatinib (ER2-positive metastatic breast cancer) [145]. Death of dormant cells induced by ferroptosis was accomplished through the repression of glutathione peroxidase 4 (GPX4) [146]. In addition to that, given the diagnostic tools that exist to date, the detection of individual dormant cells is challenging, rendering the evaluation of target efficacy almost impossible in cancer patients. Therefore, a significantly poorer prognosis and a more aggressive phenotype would be associated with dormant cells surviving these treatments. A treatment that eliminates dormant DTCs would not suffer from the same applied concerns as chronic dormancy-maintenance therapy. Patient registration in trials would be simpler because treatment would be engaged highly around the time of surgery, when systemic neoadjuvant or adjuvant therapies are already run.

### 4.4. Advances in Targeting DTCs

Therapeutic targeting of dormant DTCs has therefore attracted significant interest and constitutes a promising avenue for improved clinical outcomes of cancer patients. Unfortunately, due to the immense amount of information missing regarding cellular or surface markers that could lead to the identification of DTCs as well as limitations in current diagnostic technologies, this therapeutic strategy is still very challenging [147]. A permanent state of dormancy and therefore blocking of metastases might be achieved by interfering with the balance between ERK and p38 status, the alteration of which links to various networks and dormancy escape-promoting signaling cascades (Figure 2) [36]. The balance of p38/ERK could be altered through the functions of integrins and fibronectin, the binding of which is mediated by periostin, tenascin, macrophages or other ECM components, such as cytokines and growth factors, leading to DTCs re-awakening (Figure 3). Proteinase enzymes, secreted from neutrophils, lead to ECM remodeling and signaling integration, leading to DTC re-awakening [147].

Cellular dormancy can be studied in vitro owing to the development of a platform using a method known as ‘cancer-tissue originated spheroid method (CTOS)’, which allows for hypoxic conditions to be applied for a minimum of 7 days without requiring stimulation by growth factors [43]. Novel techniques have led to discoveries regarding the localization of dormant cells in different organ tissues [148]. For instance, dormant, prostate cancer cells, acute lymphoblastic leukemia cells [149] and multiple myeloma cells [114] tend to be found at the endosteal surface of the bone, whereas other types, such as breast and brain cancer cells, exhibit preference over the perivascular regions of organs [43,68]. It is now possible for specific stages in the life cycle of dormant cancer cells to be targeted through advances in RNA sequencing and intraviral imaging, allowing for treatments to prevent or diminish metastatic outgrowth [150]. In general, targeting large and highly vascularized tumors confers many disadvantages, including poorer treatment prognosis and, thus, it is recommended to target tumors during the earlier stages of quiescence instead [151]. Another approach for the prevention of metastatic relapse would be the implementation of studies focused in inhibiting glycolysis, hypoxia or other dormancy-promoting conditions in order to investigate their effect on dormant cell viability [152]. The identification of distinct gene expression signatures found in dormant cells could provide novel insights for discovering novel targeted therapies. However, macrophages often express molecules that are also expressed in the surface of dormant cells (e.g., VCAM and AXL) and thus, their targeting could lead to immunity interference and an ineffective or non-targeted treatment [50]. This could potentially be overcome through the concept of immunocloaking and the discovery of mechanisms that could reverse the action of disguised cells prior to their elimination [89].

### 4.5. Future Therapeutic Interventions

Numerous studies have tried to classify dormant tumor cells using biomarkers that correlated with a dormant cell state, such as protein Ki67 [153] and the apoptotic markers TUNEL and M30 or with a dormancy regulating/inducing signaling pathway, such as the TGF-β family, the PI_3_K/AKT and p38 expression [154,155]. However, such biomarkers are not only marking dormant cells but may also include other non-proliferating cell populations. Therefore, single cell-omics approaches are becoming particularly informative in delineating these cell type specific mechanisms. In an effort to develop more complex gene signatures for dormant tumor cell identification, a genome-wide study using patient-derived xenograft models glioblastoma, osteosarcoma liposarcoma and breast cancer led to the identification of a consensus expression signature differentiating dormant from fast-growing angiogenic tumors [154]. Another study using single-cell mRNA sequencing in pancreatic cancer revealed a plethora of pathways essential to preserve dormancy in pancreatic cancer-derived liver metastasis [156]. Computational and bioinformatic procedures putting together numerous dormant tumor cell biomarker studies will be crucial to outline dormant tumor cell signatures that are more generally relevant [34,157]. As already described above, a plethora of extracellular and microenvironmental signals, influence the dormant tumor cell state. Important pathways and microenvironmental causes are already widely studied, aiming to shed light on the process of determining dormant tumor cell fate. Further investigation is necessary to reveal the exact mechanisms that separate the induction of different non-proliferating tumor cell states. Such an understanding will help us identify potential dormant tumor cell treatment options to prevent (late) recurrences. A wider understanding of the signals that trigger dormancy-related therapy resistance can subsequently add to the development of current strategies toward improved treatments that prevent cancer relapse [19].

The acceptance and understanding of the mechanisms that drive colony formation during cancer metastasis is fundamental. New mechanistic studies can be designed in order to look forward and produce new advanced models appropriate to study the development of metastasis from early, genomic immature DCCs [158]. Naume and colleagues managed to identify DCCs as a surrogate marker for adjuvant treatment effects. Such studies must be linked to other therapeutic attempts, such as omics analysis of DCCs and the evaluation of the pre-therapeutic and post-therapeutic microenvironment [159]. Through such combinations, we will gain extensive insights for the development of drugs that prevent activation of dormant DCCs, target slow-growing DCCs, evolving or stimulated DCCs and explosively growing DCCs. Thus, the role of the microenvironment in metastatic colony formation and growth is of paramount importance and should inspire the search for and clinical testing of drugs that either kill DCCs without generating inflammation, senescence and without withdrawing vital stimuli provided by the microenvironment [158]. The implementation of such combinatorial methods remains a major challenge, which can become even greater when it comes to testing these novel metastasis prevention models in the clinic. The available methods to detect DCCs are currently not sensitive enough to use eradication of DCCs as a readout, making it very difficult and time consuming to perform follow-up studies. Consistent discovery of early systemic cancer, molecular risk assessment and identification of predictive markers during the invisible phase of metastatic progression should become an intensified research field in the near future. Using good substitute end-points, the implementation of adjuvant therapy studies, delivered from trial designs used for patients with evident metastasis, may become more feasible and pave the way for balanced advances of adjuvant therapies [158].

Overall, dormant tumor cells are characterized as promising targets for the prevention of metastasis. The classification of different growth-arrested cell populations is not clear, and future research is necessary to separate their phenotypes. Conventional therapies do not target dormant tumor cells, thus there is an urgent need to combine the available conventional therapies or study if higher dosages of these available treatments may eliminate dormant tumor cells [19]. A spin in drug penetration research, a major challenge in cancer treatment, is necessary to give escalation to next-generation therapies. Modern nanotechnological methodologies may accomplish this necessity. The in vivo validation of novel therapies is important to approve feasibility of dormant tumor cell eradication by nanomedicine strategies. As tumor masses consist of highly heterogeneous cell populations, conventional therapies can possibly be supplemented with innovative dormant tumor cell-targeted probes [19].

## 5. Conclusions and Future Perspectives

Cancer recurrence and metastasis constitutes a significant challenge in clinical oncology due to the ineffectiveness of treatment during later stages, eliciting a negative impact on patients’ lives. Relapse prevention is often accomplished using adjuvant therapy and by targeting residual disease; however, not all patients are benefited by these approaches. Tumor recurrence and metastasis after an extended period post-treatment ascribe to ‘tumor dormancy’, a phenomenon involving the presence of ‘disseminating tumor cells’ and their survival in a prolonged dormant state [54]. During the earlier stages of cancer, DTCs usually occupy distant niches and embed their target secondary organs prior to their affiliation with disease relapse. Tumor dormancy has been demonstrated in several cancer types, including, leukemia, multiple myeloma, melanoma, prostate, lung, breast, kidney and colon cancers [3]. It represents a significant phase in cancer development, often called as the rate limiting step during tumor progression, during which metastatic growth is not detected in the clinical context, due to the presence of low numbers of dormant tumor cells. Both cellular dormancy and tumor mass dormancy can be included in the term, with the former signifying a viable, reversible and non-proliferative cell status and the latter referring to the survival of neoplastic masses that lack the ability of progression (Figure 1) [151]. Cellular dormancy corresponds to the reversible termination of the cancer cell cycle and the entrance into a quiescent state through a G0-G1 mitotic arrest [114].

Tumor microenvironment mechanisms, such as the immune system or hypoxia, can drive tumor mass dormancy by balancing cell growth and cell apoptotic rates in a way which allows for equilibrium to be reached (Figure 1 and Figure 2) [68]. The equilibrium allowing for growth arrest and maintenance of dormant cancer masses is controlled by anti- and pro-angiogenic factors induced by the tumor microenvironment [160]. Overexpression of cell cycle inhibitors, such as p27 and p21, could prevent further cell expansion by arresting tumor cell growth and therefore initiating cell quiescence and tumor dormancy [54]. Different, non-mutually exclusive but complementary subdivisions of tumor dormancy exist. First and foremost, cellular dormancy refers to quiescent entry of either small groups or solitary cells (Figure 1), governed by intrinsic or extrinsic mechanisms, providing a therapeutic window for anti-cancer therapies [24]. The stability between cell apoptosis and division caused by poor vascularization drives angiogenic dormancy. Immunologic dormancy refers to immune evasion and the maintenance of a proliferating tumor mass through the cytotoxic effects secreted from the immune system [18].

There is a perplexing inconsistency between the lack of completely malignant DCCs during a surgery performed in order to remove a primary tumor, their less frequent detection compared to the extremely rare ‘immature’ DCCs, and the apparent genomic parallel between some primary tumors and obvious metastasis. Thus, there is a question mark whether metastasis originates by early DCCs or late DCCs. Early DCCs need to evolve and obtain significant changes at the distant site, whereas late DCCs can find metastasis after adapting to the new environment followed by clonal development. Studies in animal models suggest that a percentage of 80% of metastasis cases derived from early DCCs, while on the other hand genomic analysis of DCC together with the primary tumor and metastasis has showed contradicting conclusions. When DCCs are isolated from patients without obvious metastasis, their genomic characteristics reveal an early genomic departure from the key clone of the primary tumor, indicating dissemination early in ‘genomic’ time [9,112,161,162,163,164,165,166]. On the other hand, qualified sequencing of primary tumors and their metastatic potential showed that late genomic dissemination of metastasis founder cells is about 60% of the cases tested [167,168,169]. By putting all these findings together, it has been suggested that development of metastases could be multiclonal and may be generated by several waves of seeding [161,170,171]. Early DCCs may prepare the niche for late DCCs, with clonal to clonal support being significant for the clinical appearance of metastases [158].

In this article, we reviewed the signaling pathways, the intrinsic and extrinsic mechanisms that either sustain dormancy, making DTCs more challenging to identify and eradicate, or enable the escape of these cells from their dormant state at distal sites, causing metastasis. Most of the known signals and mechanisms are characterized as organ-specific, suggesting that dormancy is regulated through the tissue microenvironment and that pharmacological targeting of cancer cell dormancy might involve a highly personalized approach. Thus, targeting DTCs has been increasingly gaining attention. DTCs self-seed for prolonged periods of time at their dormant state, giving scientists the opportunity for therapeutic intervention and experimentation. A promising approach for the complete eradication of DTCs is by targeting both proliferating and dormant cells by using a combination of drugs that will also prevent cancer recurrence. Remarkably, several studies support the sequential combination of drugs rather than co-treatment. There is an obvious need for improving preclinical dormancy models and developing sensitive diagnostic imaging tools that can detect dormant cancer cells. Finally, designing new dormancy-targeting therapeutic drugs to overcome resistance to metastatic cancer therapies is urgently needed.

## Figures and Tables

**Figure 1 ijms-23-13931-f001:**
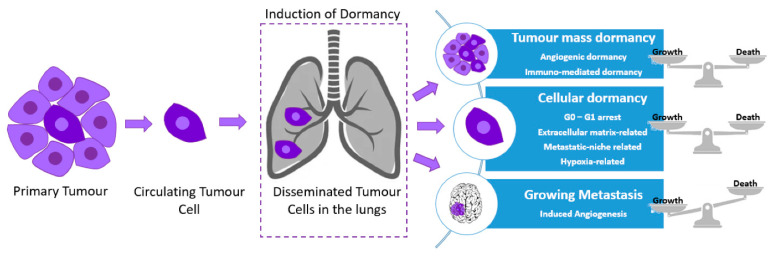
Cellular dormancy and tumor mass dormancy in the lungs. Tumor cells leave the primary tumor and enter the blood circulation (circulating tumor cells (CTCs)). These aggressive cancer cells occupy distant organ sites where they self-seed and become dormant (disseminated tumor cells (DTCs)). DTCs can develop a secondary tumor in a distant organ site immediately or at a later stage (a process known as metastasis), or they can maintain the dormant state for prolonged periods of time. Over the years, these dormant tumor cells can escape from the dormancy state and develop metastasis. When the balance between growth and death is achieved, the cells/tumor mass are/is in cell/tumor mass dormancy. In cellular dormancy, growth and death rates are limited.

**Figure 2 ijms-23-13931-f002:**
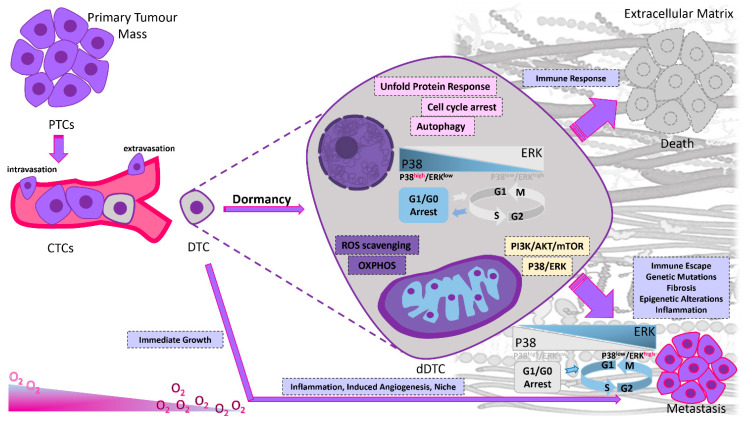
Dormancy in cancer progression. Primary tumor cells can leave the primary tumor site and enter the blood circulation as circulating tumor cells (CTCs). When CTCs reach a target organ, they seed into the new tissue as DTCs. Their fate is diverse depending on their cell autonomous intracellular signals and due to the local environmental signals. If DTCs fail to adapt to the new microenvironment they die due to different immune responses. If the local microenvironment can provide missing complementary cues to the DTCs, they can resume their proliferative properties. When DTCs are challenging a hostile environment may enter a dormant state due to the interactions of extracellular factors (mainly hypoxia and immune cytotoxicity) and intracellular pathways. In response to signals that are not fully elucidated but appear to involve modifications of tumor microenvironment, such as extracellular matrix remodeling, fibrosis, neovascularization, epigenetic alterations, chronic inflammation, immune escape, genetic mutations and tissue specific mechanisms in each metastatic site, tumor cells escape from dormancy, start to proliferate and ultimately form macroscopic metastases in target organs. Proliferating DTCs have to evade the immune system; otherwise, the immune system will keep them in check by preventing their expansion. The balance between activated extracellular signal-regulated kinases (ERK1/2) and activated p38α/β has been correlated to cell dormancy in vitro and in vivo. The ERK/p38 ratio is indicative of the dormant phenotype; a high ratio induces tumor growth, whereas a low ratio promotes tumor dormancy. P38 drives a downstream program that resembles endoplasmic reticulum (ER) stress and coordinates a stress-related transcriptional program. Activation of the p38 MAPK in dormant cells induces the unfolded-protein response (UPR). Reduced PI_3_K/AKT signaling due to extracellular regulation as a response to environmental signals plays a key role in switching between cell proliferation and dormancy. Quiescent cancer cells mainly depend on OXPHOS and lipid oxidation for their survival. Specific molecular pathways regulate the metabolic shift toward OXPHOS in these cells. Despite the activation of glycolysis or OXPHOS, the AMPK stress response maintains a quiescent state by repressing cellular proliferation through mTOR inhibition. Autophagy sustains the acquisition of a dormant phenotype by providing appropriate energy balance under metabolic stress and by managing the ROS accumulation.

**Figure 3 ijms-23-13931-f003:**
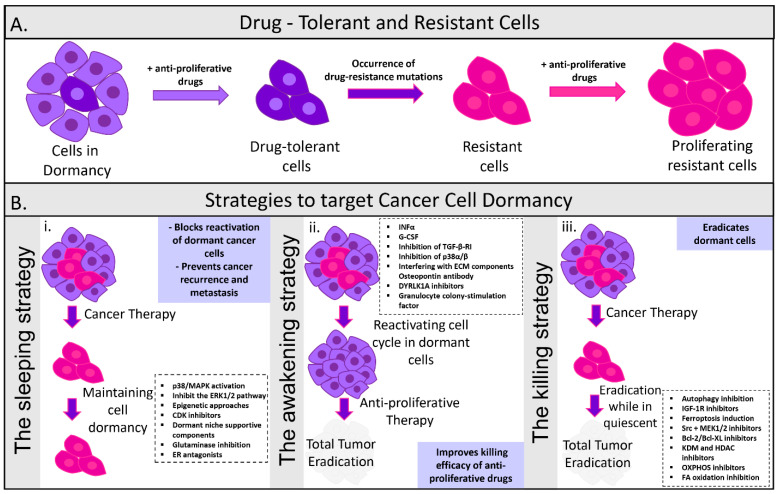
Strategies to target cancer cell dormancy. (**A**) Dormancy is identified as a key feature of cells surviving their first exposure to anti-proliferative cancer drugs. Drug tolerant cells are able to escape the highly selective pressure of anti-proliferative agents and become resistant. Although drug tolerance is a transient non mutational phenotype, rare drug-resistant mutations occur. Drug tolerant persisted cells resume normal growth even in the presence of a drug and form colonies. These drug-resistant cancer cell colonies can proliferate even in the presence of anti-proliferative drugs. (**B**) Overview of the current pharmacological strategies that aim to (**i**) maintain cancer cells in the dormant state, (**ii**) reactivate dormant cells to increase their susceptibility to anti-proliferative drugs, and (**iii**) eradicate dormant cancer cells while in quiescent. The maintenance of cancer cells in a dormant state can be achieved by suppressing proliferative signaling, activating dormant pathways, or delivering components of the dormant niches. The reactivation of the dormant cells is reached by targeting dormancy-promoting components secreted by the microenvironment in order to force dormant cells into the cell cycle. The elimination of the dormant cancer cells can be accomplished by designing drugs that will directly kill disseminating tumor cells.

## Data Availability

Not applicable.

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
