# Peer review of "Regulation of Metastatic Tumor Dormancy and Emerging Opportunities for Therapeutic Intervention"

_ijms, 2022, doi:10.3390/ijms232213931_

Round 1

Reviewer 1 Report

In a nice appreciable effort, the authors present a broad overview of the dormancy phenomenon and associated therapeutic opportunities. 

(A) However, the manuscript would benefit from better organization and proper text sharpening to overcome the frequent lack of consistency in many settings.  For instance: 

1. In the abstract (line 27): the authors talk hinted on self-seeding, an independent concept from dormancy, without a clear link to the review’s subject or further expanding in the main text. 

2. The introduction navigates from one idea to another without smooth transitions. Perhaps the authors can consider shortening this section and integrate these different notions within the main text. 

3. There is a level of redundancy between different sections in the manuscript. For specific examples: the definition of different types of dormancies, the effect of specific factors or pathways on dormancy dynamics (TSP1, uPAR, p38, ECM), and many others. Perhaps one suggestion would be to merge the “3. Escape from Dormancy” section with the “2.Mechanisms of Dormancy Induction and Maintenance” and the text can be shortened accordingly. Another suggestion would be synthesizing the literature focused on one factor or family of factors (i.e. IL6 and IGFs) in one setting where the net effect of such factors on dormancy is clear.  

4. Related to the previous points, authors should consider shortening the conclusion section as it includes redundant text. One paragraph should suffice. 

5. A suggestion for the immune system subsection: categorizing the text under innate and acquired immunity might streamline the narrative. 

6. This is not mandatory, but it would be informative if the authors provided a figure or table where the dormancy regulatory pathway/factor is matched to the lineage it was investigated in in addition to the net effect on metastasis. 

(B) In our view, a major limitation of the current version of the manuscript is that the authors often don’t specify the organ where the cancer cells dormancy was investigated. Whether they mean dormancy at the primary tumor setting or metastatic is not clear. Since the focus of this manuscript is the latter, it would be informative to include the metastatic organ on summarizing the different studies’ results. 

(C) Some parts of the manuscripts are vague and poorly informative. For instance, line 321, the authors talk about miRNAs but don’t give specific examples of these miRNAs and their net effect on dormancy dynamics. Another example is discussing GIST (line 216). It is not clear whether these experiments were done in a metastatic setting and how this informs on the behavior of dormant DTCs. These sections should be considered to be removed to streamline the manuscript.

(D) References: some claims are not mirrored with supporting references such as line 70-71, 663-665. Alternatively, some references are incomplete (ex: line 806). In some settings, the authors cite review articles rather than the actual studies that led to the conclusion (ex: line 653-655). 

(E) It might be beneficial if the concept of early vs late seeding is included briefly in the discussion. The Christoph Klein group published some reviews that might be of guidance. This is relevant from both mechanistic and therapeutic aspects, which the authors focus on. 

Author Response

Response to Reviewer 1

[General Comment] In a nice appreciable effort, the authors present a broad overview of the dormancy phenomenon and associated therapeutic opportunities.

Response: Thank you very much for appreciating our efforts to prepare this manuscript. We have carefully read your constructive comments and tried our best to address them all one-by-one. We believe that the manuscript has been improved towards IJMS standards upon this revision.

[Comment A] (A) However, the manuscript would benefit from better organization and proper text sharpening to overcome the frequent lack of consistency in many settings.  For instance:

  1. In the abstract (line 27): the authors talk hinted on self-seeding, an independent concept from dormancy, without a clear link to the review’s subject or further expanding in the main text.

Response: Thank you very much for pointing this out. We have now improved the overall organization of the manuscript and revised the text to be more concise in different parts of the paper. We have also aimed to maintain consistency and remove redundancy in various sections of the manuscript.

More specifically, we have revised the indicated sentences as follows:

Page 1, Lines 27-32: Studies on the ability of the metastatic cancer cells to cease proliferation and survive in a quiescent state before re-initiating proliferation and colonization years after successful treatment, will pave the way toward developing innovative therapeutic strategies against dormancy-mediated metastatic outgrowth.

  1. The introduction navigates from one idea to another without smooth transitions. Perhaps the authors can consider shortening this section and integrate these different notions within the main text.

Response: Thank you for your comment. We have now shortened this section, by removing the following sections, to enhance smooth transition between concepts.

Page 1, Lines 38-44

Page 2, Lines 51-52

Page 2, Lines 57-59

Page 2, Lines 77-82

Page 3, Lines 98-99

Page 3, Lines 102-105

  1. There is a level of redundancy between different sections in the manuscript. For specific examples: the definition of different types of dormancies, the effect of specific factors or pathways on dormancy dynamics (TSP1, uPAR, p38, ECM), and many others. Perhaps one suggestion would be to merge the “3. Escape from Dormancy” section with the “2. Mechanisms of Dormancy Induction and Maintenance” and the text can be shortened accordingly. Another suggestion would be synthesizing the literature focused on one factor or family of factors (i.e. IL6 and IGFs) in one setting where the net effect of such factors on dormancy is clear.

Response: Thank you for your comment. In a way, we understand why you suggest this, but we prefer to keep the “Escape from Dormancy” section and the “Mechanisms of Dormancy Induction and Maintenance” section of the article separately, since we want the two concepts to be clear, so that it would be easier for the reader to understand and separate the mechanisms that we encounter in the two processes.

  1. Related to the previous points, authors should consider shortening the conclusion section as it includes redundant text. One paragraph should suffice.

Response: Thank you for your comment. We have shortened the conclusion section by removing lines 1154-1190 in pages 24-25.

  1. A suggestion for the immune system subsection: categorizing the text under innate and acquired immunity might streamline the narrative.

Response: Thank you for this suggestion. This section has been improved and separated into innate and adaptive immunity. Please see Pages 10-11, Lines 484-575.

  1. This is not mandatory, but it would be informative if the authors provided a figure or table where the dormancy regulatory pathway/factor is matched to the lineage it was investigated in in addition to the net effect on metastasis.

Response: Thank you for this interesting comment. To address this and the next comment, we have now included a table (table 1) which includes the requested information. Please see below for more details.

[Comment B] (B) In our view, a major limitation of the current version of the manuscript is that the authors often don’t specify the organ where the cancer cells dormancy was investigated. Whether they mean dormancy at the primary tumor setting or metastatic is not clear. Since the focus of this manuscript is the latter, it would be informative to include the metastatic organ on summarizing the different studies’ results.

Response to Comment A6 and B: Thank you for pointing this out. In order to match the factor to the lineage it was investigated in addition to the net of metastasis and the organ where the cancer cells dormancy was investigated, we created a table (table 1, page 17) where we included different factors implicated in these processes, the associated mechanisms that sustain dormancy or promote the escape from dormancy, the tumor type, the respective metastatic site and the different studies that support these findings. Please see Page 17, Lines 811-831.

[Comment C] (C) Some parts of the manuscripts are vague and poorly informative. For instance, line 321, the authors talk about miRNAs but don’t give specific examples of these miRNAs and their net effect on dormancy dynamics. Another example is discussing GIST (line 216). It is not clear whether these experiments were done in a metastatic setting and how this informs on the behavior of dormant DTCs. These sections should be considered to be removed to streamline the manuscript.

Response: Thank you very much for your feedback regarding these vague parts of the manuscript. We have now removed these sections in order to simplify and better streamline the manuscript.

[Comment D] (D) References: some claims are not mirrored with supporting references such as line 70-71, 663-665. Alternatively, some references are incomplete (ex: line 806). In some settings, the authors cite review articles rather than the actual studies that led to the conclusion (ex: line 653-655).

Response: Thank you for this suggestion. We removed the references that did not support some of the claims and provided the following references to further support these statements.

Lines 72-73 (Previously known as lines 70-71), Page 2:

DASGUPTA, A., LIM, A. R. & GHAJAR, C. M. 2017. Circulating and disseminated tumor cells: harbingers or initiators of metastasis? Molecular Oncology, 11, 40-61.

ESLAMI-S, Z., CORTÉS-HERNÁNDEZ, L. E., THOMAS, F., PANTEL, K. & ALIX-PANABIÈRES, C. 2022. Functional analysis of circulating tumour cells: the KEY to understand the biology of the metastatic cascade. British Journal of Cancer, 127, 800-810.

Lines 833-837 (Previously known as lines 663-665), Page 17:

MEACHAM, C. E. & MORRISON, S. J. 2013. Tumour heterogeneity and cancer cell plasticity. Nature, 501, 328-37.

PUIG, I., TENBAUM, S. P., CHICOTE, I., ARQUÉS, O., MARTÍNEZ-QUINTANILLA, J., CUESTA-BORRÁS, E., RAMÍREZ, L., GONZALO, P., SOTO, A., AGUILAR, S., EGUIZABAL, C., CARATÙ, G., PRAT, A., ARGILÉS, G., LANDOLFI, S., CASANOVAS, O., SERRA, V., VILLANUEVA, A., ARROYO, A. G., TERRACCIANO, L., NUCIFORO, P., SEOANE, J., RECIO, J. A., VIVANCOS, A., DIENSTMANN, R., TABERNERO, J. & PALMER, H. G. 2018. TET2 controls chemoresistant slow-cycling cancer cell survival and tumor recurrence. J Clin Invest, 128, 3887-3905.

WOLTER, K. & ZENDER, L. 2020. Therapy-induced senescence - an induced synthetic lethality in liver cancer? Nat Rev Gastroenterol Hepatol, 17, 135-136.

Line 988 (Previously known as line 806), Page 20:

Incomplete reference fixed.

Line797-798 (Previously known as lines 653-655), Page 17:

GAO, H., CHAKRABORTY, G., LEE-LIM, A. P., MO, Q., DECKER, M., VONICA, A., SHEN, R., BROGI, E., BRIVANLOU, A. H. & GIANCOTTI, F. G. 2012. The BMP inhibitor Coco reactivates breast cancer cells at lung metastatic sites. Cell, 150, 764-79.

[Comment E] (E) It might be beneficial if the concept of early vs late seeding is included briefly in the discussion. The Christoph Klein group published some reviews that might be of guidance. This is relevant from both mechanistic and therapeutic aspects, which the authors focus on.

Response: Thank you very much for your comment, and for suggesting us reading the very interesting work by Christoph Klein and his group. We have included the concept of early vs late seeding in the discussion. Please see Pages 23-24, Lines 1116-1137.

Reviewer 2 Report

In this manuscript, Tamamouna and colleagues describe the regulation of metastatic tumor dormancy and review a therapeutic intervention. The topic is very interesting. However, there are several similar reviews for tumor dormancy and therapeutic implications, such as IJMS 2021, 22:4862. In this case, the authors could focus on the therapeutic intervention and mechanism for treating tumor dormancy and address a specific gap in the field.

 Main points:

However, the authors should provide a conclusion with evidence in each paragraph or section.

 The manuscript is organized in multiple paragraphs, which at times contain repetitive sentences and repeat content.

 The authors should improve the quality of some of the figures.

 In addition, the text has not been simplified, and the review remains challenging to read.

Author Response

Response to Reviewer 2

[General Comment] In this manuscript, Tamamouna and colleagues describe the regulation of metastatic tumor dormancy and review a therapeutic intervention. The topic is very interesting. However, there are several similar reviews for tumor dormancy and therapeutic implications, such as IJMS 2021, 22:4862. In this case, the authors could focus on the therapeutic intervention and mechanism for treating tumor dormancy and address a specific gap in the field.

Response: Thank you for your comments. We have carefully gone through your comments and tried our best to address them one by one. We are confident that the manuscript has now been significantly improved. As suggested, we have focused on the therapeutic interventions and the current gaps in the field. For this reason, we have added a new section in the manuscript “4.5. Future Therapeutic Interventions”, see Pages 21-22, Lines 1017-1079.  

Main points:

[Comment 1] However, the authors should provide a conclusion with evidence in each paragraph or section.

Response: Thank you for your insightful suggestion. We have provided a conclusion in each section of the manuscript which can be found at the following lines in the manuscript:

Page 4, Lines 180-183

Page 5, Lines 232-239

Page 7, Lines 305-311

Page 8, Lines 349-355

Page 8, Lines 387-394

Page 10, Lines 485-491

Page 15, Lines 695-703

Page 17, Lines 802-809

Page 18, Lines 878-880

Page 19, Lines 915-919

Page 20, Lines 957-961

Page 22-23, Lines 1065-1078

[Comment 2] The manuscript is organized in multiple paragraphs, which at times contain repetitive sentences and repeat content.

Response: Thank you for the comment. We have now removed some parts of the manuscript in order to keep it short and more to the point. These are the following:

Page 1, Lines 38-44

Page 2, Lines 51-52

Page 2, Lines 57-59

Page 2, Lines 77-82

Page 3, Lines 98-99

Page 3, Lines 102-105

Page 5, Lines 222-228

Pages 7-8, Lines 340-348

Pages 24-25, Lines 1153-1189

[Comment 3] The authors should improve the quality of some of the figures.

Response: Thank you for this remark. We have now improved the quality of the figures, by increasing the image resolution.

[Comment 4] In addition, the text has not been simplified, and the review remains challenging to read.

Response: Thank you for this comment. We have tried to simplify the text throughout according to your suggestion. We hope that this will improve reading flow and allow better understand the content of the manuscript.

Round 2

Reviewer 2 Report

This version of the manuscript is improved. There is no more comment.